# Learning to Infer Implicit Surfaces
# without 3D Supervision

**Shichen Liu**[†,§], **Shunsuke Saito**[†,§], **Weikai Chen (✉)**[†], **and Hao Li**[†,§,‡]

[†]USC Institute for Creative Technologies
[§]University of Southern California
[‡]Pinscreen
{liushichen95, shunsuke.saito16, chenwk891}@gmail.com   hao@hao-li.com

## Abstract

Recent advances in 3D deep learning have shown that it is possible to train highly effective deep models for 3D shape generation, directly from 2D images. This is particularly interesting since the availability of 3D models is still limited compared to the massive amount of accessible 2D images, which is invaluable for training. The representation of 3D surfaces itself is a key factor for the quality and resolution of the 3D output. While explicit representations, such as point clouds and voxels, can span a wide range of shape variations, their resolutions are often limited. Mesh-based representations are more efficient but are limited by their ability to handle varying topologies. Implicit surfaces, however, can robustly handle complex shapes, topologies, and also provide flexible resolution control. We address the fundamental problem of learning implicit surfaces for shape inference without the need of 3D supervision. Despite their advantages, it remains nontrivial to (1) formulate a differentiable connection between implicit surfaces and their 2D renderings, which is needed for image-based supervision; and (2) ensure precise geometric properties and control, such as local smoothness. In particular, sampling implicit surfaces densely is also known to be a computationally demanding and very slow operation. To this end, we propose a novel ray-based field probing technique for efficient image-to-field supervision, as well as a general geometric regularizer for implicit surfaces, which provides natural shape priors in unconstrained regions. We demonstrate the effectiveness of our framework on the task of single-view image-based 3D shape digitization and show how we outperform state-of-the-art techniques both quantitatively and qualitatively.

## 1   Introduction

The efficient learning of 3D deep generative models is the key to achieving high-quality shape reconstruction and inference algorithms. While supervised learning with direct 3D supervision has shown promising results, its modeling capabilities are constrained by the quantity and variations of available 3D datasets. In contrast, far more 2D photographs are being taken and shared over the Internet, than can ever be watched. To exploit the abundance of image datasets, various differentiable rendering techniques [1, 2, 3, 4] were introduced recently, to learn 3D generative models directly from massive amounts of 2D pictures. While several types of shape representations have been adopted, most techniques are based on explicit surfaces, which often leads to poor visual quality due to limited resolutions (e.g., point clouds, voxels) or fail to handle arbitrary topologies (e.g., polygonal meshes).

Implicit surfaces, on the other hand, describe a 3D shape using an iso-surface of an implicit field and can therefore handle arbitrary topologies, as well as support multi-resolution control to ensure high-fidelity modeling. As demonstrated by several recent 3D supervised learning methods [5, 6, 7, 8],

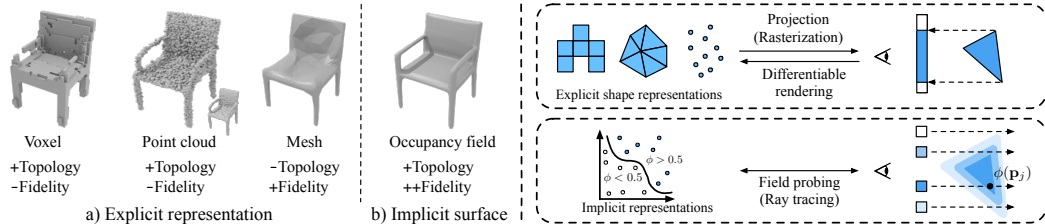

Figure 1: While explicit shape representations may suffer from poor visual quality due to limited resolutions or fail to handle arbitrary topologies (a), implicit surfaces handle arbitrary topologies with high resolutions in a memory efficient manner (b). However, in contrast to the explicit representations, it is not feasible to directly project an implicit field onto a 2D domain via perspective transformation. Thus, we introduce a field probing approach based on efficient ray sampling that enables unsupervised learning of implicit surfaces from image-based supervision.

implicit representations are particularly advantageous over explicit ones, and naturally encode a 3D surface at infinite resolution with minimal memory footprint.

Despite these benefits, it remains challenging to achieve unsupervised learning of implicit surfaces only from 2D images. First, it is non-trivial to relate the changes of the implicit surface with that of the observed images. An explicit surface, on the other hand, can be easily projected and shaded onto an image plane (Figure 1 right). By inverting such process, one can obtain gradient flows that supervise the generation of the 3D shape. However, it is infeasible to directly project an implicit field onto a 2D domain via transformation. Instead, rendering an implicit surface relies on ray sampling techniques to densely evaluate the field, which may lead to very high computational cost, especially for objects with thin structures. Second, it is challenging to ensure precise geometric properties such as local smoothness of an implicit surface. This is critical to generating plausible shapes in unconstrained regions, especially when only image-based supervision is available. Unlike mesh-based surface representations, it is not straightforward to obtain geometric properties, e.g. normal, curvature, etc., for an implicit surface, as the shape is implicitly encoded as the level set of a scalar field.

We address the above challenges and propose the first framework for learning implicit surfaces with only 2D supervision. In contrast to 3D supervised learning, where a signed distance field can be computed from the 3D training data, 2D images can only provide supervision on the binary occupancy of the field. Hence, we formulate the unsupervised learning of implicit fields as a classification problem such that the *occupancy probability* at an arbitrary 3D point can be predicted. The key to our approach is a novel *field probing* approach based on efficient ray sampling that achieves image-to-field supervision. Unlike conventional sampling methods [9], which excessively cast rays passing through all image pixels and apply binary search along the ray to detect the surface boundary, we propose a much more efficient approach by leveraging sparse sets of *3D anchor points* and *rays*. In particular, the anchor points probe the field by evaluating the occupancy probability at its location, while the rays aggregate the information from the anchor points that it intersects with. We assign a spherical supporting region to each anchor point to enable the ray-point intersection. To further improve the boundary modeling accuracy, we apply importance sampling in both 2D and 3D space to allocate more rays and anchor points around the image and surface boundaries respectively.

While geometric regularization for implicit fields is largely unexplored, we propose a new method for constraining geometric properties of an implicit surface using the approximated derivatives of the field with a finite difference method. Since we only care about the decision boundary of the field, regularizing the entire 3D space would introduce scarcity of constraints in the region of interest. Hence, we further propose an importance weighting technique to draw more attention to the surface region. We validate our approach on the task of single-view surface reconstruction. Experimental results demonstrate the superiority of our method over state-of-the-art unsupervised 3D deep learning techniques, that are based on alternative shape representations, in terms of quantitative and qualitative measures. Comprehensive ablation studies also verify the efficacy of proposed probing-based sampling technique and the implicit geometric regularization.

Our contributions can be summarized as follows: (1) the first framework that enables learning of implicit surfaces for shape modeling without 3D supervision; (2) a novel field probing approach based on anchor points and probing rays that efficiently correlates the implicit field and the observed images;

(3) an efficient point and ray sampling method for implicit surface generation from image-based supervision; (4) a general formulation of geometric regularization that can constrain the geometric properties of a continuous implicit surface.

## 2   Related Work

**Geometric Representation for 3D Deep Learning.**   A 3D surface can be represented either *explicitly* or *implicitly*. Explicit representations mainly consist of three categories: voxel-, point- and mesh-based. Due to their uniform spatial structures, voxel-based representations [10, 11, 12, 13] have been extensively explored to replicate the success of 2D convolutional networks onto the 3D regular domain. Such volumetric representations can be easily generalized across shape topologies, but are often restricted to low resolutions due to large memory requirements. Progress has also been made in reconstructing point clouds from single images using point feature learning [14, 15, 16, 17, 3]. While being able to describe arbitrary topologies, point-based representations are also restricted by their resolution capabilities since dense samples are needed. Mesh representations can be more efficient since they naturally describe mesh connectivity and are hence, suitable for 2-manifold representations. Recent advances have focused on reconstructing mesh geometry from point clouds [18] or even a single image [19]. AtlasNet [18] learns an implicit representation that maps and assembles 2D squares to 3D surface patches. Despite the compactness of mesh representations, it remains challenging to modify the vertex connections, making it unsuitable for modeling shapes with arbitrary topology.

Unlike explicit surfaces, implicit surface representations [20, 21] depict a 3D shape by extracting the iso-surface from a continuous field. For implicit surfaces, a generative model can have more flexibility and expressiveness for capturing complex topologies. Furthermore, multi-resolution representations and control enable them to also capture fine geometric details at *arbitrary* resolution and also reduce the memory footprint during training. Recent works [22, 5, 6, 7, 8, 23] have shown promising results on supervised learning for 3D shape inference based on implicit representations. Our approach further pushes the envelope by achieving 3D-unsupervised learning of implicit generative shape modeling solely from 2D images.

**Learning Shapes from 2D Supervision.**   Training a generative model for 3D shapes typically requires direct 3D supervision from a large corpus of shape collections [10]. However, 3D model databases are still limited compared to the massive availability of 2D photos, especially since acquiring clean and high-fidelity ground-truth 3D models still requires a tedious 3D capture process [24, 25]. A number of techniques have been introduced to exploit 2D training data to overcome this limitation, and use key points [26], silhouettes [4, 1, 2, 27], and shading cues [28] for supervision. In particular, Yan et al. [4] obtain the shape supervision by measuring the loss between the perspectively transformed volumes with the ground-truth silhouettes. To achieve even denser 2D supervision, differentiable rendering (DR) techniques have been proposed to relate the changes in the observed pixels with that of the 3D models. One line of DR research focuses on differentiating the rasterization-based rendering. Loper and Black [29] introduce an approximate differentiable renderer that generates rendering derivatives. Kato et al. [1] achieve single-view mesh reconstruction using a hand-crafted function to approximate the gradient of mesh rendering. Liu et al. [2] instead propose to render meshes with differentiable functions to obtain the gradient. In addition to polygon meshes, Insafutdinov et al. [3] propose the use of differentiable point clouds to learn shapes and poses in an unsupervised manner. Another direction of DR work aims to differentiate the ray tracing procedure during rendering. Li et al. [30] introduce a differentiable ray tracer through edge sampling. Aside from silhouettes, shading and appearances in image space also provides supervision cues for learning fine-grained shape representations in category specific domains such as 3D face reconstruction [31, 32, 33, 34, 35] and material inference [36, 37, 38]. Whereas existing methods focus on learning shapes from 2D supervisions and the use of explicit shape representations (i.e., voxels, point clouds, and meshes), we present the first framework for unsupervised learning of implicit surface representations by differentiating the implicit field rendering. With our framework, one can reconstruct shapes with arbitrary topology at arbitrary resolution from a single image without requiring any 3D supervision.

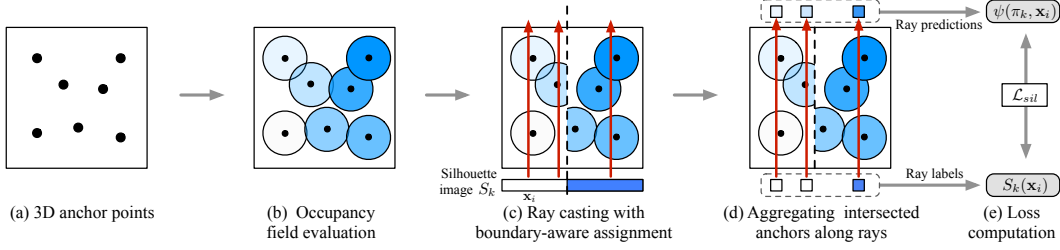

(a) 3D anchor points    (b) Occupancy field evaluation    (c) Ray casting with boundary-aware assignment    (d) Aggregating intersected anchors along rays    (e) Loss computation

Figure 2: Ray-based field probing technique. (a) A sparse set of 3D anchor points are distributed to sense the field by sampling the occupancy value at its location. (b) Each anchor is assigned a spherical supporting region to enable ray-point intersection. The anchor points that have higher probability to stay inside the object surface are marked with deeper blue. (c) Rays are cast passing through the sampling points $\{\mathbf{x}_i\}$ on the 2D silhouette under the camera views $\{\pi_k\}$ (blue indicates object interior and white otherwise). (d) By aggregating the information from the intersected anchor points via max pooling, one can obtain the prediction for each ray. (e) The silhouette loss is obtained by comparing the prediction with the ground-truth label in the image space.

# 3 Unsupervised Learning of Implicit Surfaces

**Overview.** Our goal is to learn a generative model for implicit surfaces that infers 3D shapes solely from 2D images. Unlike direct supervision with 3D ground truth, which supports the computation of a continuous signed distance field with respect to the surface, 2D observations can only provide guidance on the occupancy of the implicit field. Hence, we formulate the unsupervised learning of implicit surfaces as a classification problem. Given $\{I_k\}_{k=1}^{N_K}$ images of an object $O$ from different views $\{\pi_k\}_{k=1}^{N_K}$ as supervision signals, we train a neural network that takes a single image $I_k$ and produce a continuous occupancy probability field, whose iso-surface at 0.5 depicts the shape of $O$. Our pipeline is based on a novel ray-based field probing technique as illustrated in Figure 2. Instead of excessively casting rays to detect the surface boundary, we probe the field using a sparse set of 3D anchor points and rays. The anchor points sense the field by sampling the occupancy probability at its location, and are assigned a spherical supporting region to ease the computation of ray-point intersection. We then correlate the field and the observed images by casting the probing rays, which originate from the viewpoint and pass through the sampling points of the images. The ray, that passes through the image pixel $\mathbf{x}_i$, given the camera parameter $\pi_k$, obtains its prediction $\psi(\pi_k, \mathbf{x}_i)$ by aggregating the occupancy values from the anchor points whose supporting regions intersect with it. By comparing $\psi(\pi_k, \mathbf{x}_i)$ with the ground-truth label of $\mathbf{x}_i$, we can obtain error signals that supervise the generation of implicit fields. Note that when detecting ray-point intersections, we apply a boundary-aware assignment to remove ambiguity, which is detailed in Section 3.1.

**Network Architecture.** We demonstrate our network architecture in Figure 3. Following the recent advances in unsupervised shape learning [4, 1], we use 2D silhouettes of the objects as the supervision for network training. Our framework consist of two components: (1) an image encoder $g$ that maps the input image $I$ to a latent feature $\mathbf{z}$; and (2) an implicit decoder $f$ that consumes $\mathbf{z}$ and a 3D query point $\mathbf{p}_j$ and infers its occupancy probability $\phi(\mathbf{p}_j)$. Note that the implicit decoder generates a continuous prediction ranging from 0 to 1, where the estimated surface can be extracted at the decision boundary of 0.5 (Figure 3 right).

## 3.1 Sampling-Based 2D Supervision

To compute the prediction loss of the implicit decoder, a key step is to properly aggregate the information collected throughout the field probing process for each ray. Given a continuous occupancy field and a set of anchor points along a ray $\mathbf{r}$, the probability that $\mathbf{r}$ hits the object interior can be considered as an aggregation function:

$$\psi\left(\pi_k, \mathbf{x}_i\right) = \mathcal{G}\left(\left\{\phi\left(\mathbf{c} + \mathbf{r}\left(\pi_k, \mathbf{x}_i\right) \cdot t_j\right)\right\}_{j=1}^{N_p}\right), \tag{1}$$

where $\mathbf{r}(\pi_k, \mathbf{x}_i)$ denotes the ray direction that intersects with the image pixel $\mathbf{x}_i$ in the viewing direction $\pi_k$; $\mathbf{c}$ is the camera location; $N_p$ is the number of 3D anchor points; $t_j$ indicates the sampled location along the ray for each anchor point; $\phi(\cdot)$ is the occupancy function that returns the occupancy probability of the input point; $\psi$ denotes the predicted occupancy for ray $\mathbf{r}(\pi_k, \mathbf{x}_i)$. Since whether the ray $\mathbf{r}$ hits the object interior is determined by the maximum occupancy value detected along the ray, in this work, we adopt $\mathcal{G}$ as a max-pooling operation due to its computational efficiency and effectiveness demonstrated in [4]. By considering the $l_2$ differences between the predictions and the ground-truth silhouette, we can obtain the silhouette loss $\mathcal{L}_{sil}$:

$$\mathcal{L}_{sil} = \frac{1}{N_r} \sum_{i=1}^{N_r} \sum_{k=1}^{N_K} \|\psi(\pi_k, \mathbf{x}_i) - S_k(\mathbf{x}_i)\|^2, \tag{2}$$

where $S_k(\mathbf{x}_i)$ is a bilinearly interpolated silhouette at $\mathbf{x}_i$ under the $k$-th viewpoint; $N_r$ and $N_K$ denote the number of 2D sampling points and camera views, respectively.

**Boundary-Aware Assignment.** To facilitate the computation of ray-point intersections, we model each anchor point as a sphere with a non-zero radius. While such a strategy works well in most cases, erroneous labeling may occur in the vicinity of the decision boundary. For instance, a ray that has no intersection with the target object may still have a chance to hit the supporting region of an anchor point whose center lies inside the object. Since we use max-pooling as the aggregating function, the ray may be wrongly labeled as an intersecting ray. To resolve this issue, we use 2D silhouettes as additional prior by filtering out the anchor points on the wrong side. In particular, if a ray is passing through a pixel belonging to the inside/outside of the silhouette, the anchor points lying outside/inside of the 3D object are ignored when detecting intersections (Figure 2 (c)). This boundary-aware assignment can significantly improve the quality and reconstructed details, which is demonstrated in the ablation study in Section 4.

**Importance Sampling.** A naive solution for distributing anchor points and probing rays is to apply random sampling. However, as the occupancy of the target object may be highly sparse over the 3D space, random sampling could be extremely inefficient. We propose an importance sampling approach based on shape cues obtained from the 2D images for efficient sampling of rays and anchor points. The main idea is to draw more samples around the surface boundary, which is equivalent to the 2D contour of the object in image space. For ray sampling, we first obtain the contour map $W_r(\mathbf{x})$ by applying Laplacian operator over the input silhouette. We then generate a Gaussian mixture distribution by positioning the individual kernels to each pixel of $W_r(\mathbf{x})$ and setting the kernel height as the pixel intensity at its location. The rays are then generated by sampling from the resulting distribution. Similarly, to generate the 3D contour map $W_p(\mathbf{p})$, we apply mean filtering to the 3D visual hulls computed from the multi-view silhouettes. The anchor points are then sampled from a 3D Gaussian mixture distribution model created in a similar fashion to the 2D case, which yields the probabilistic density function of the sampling as:

$$\begin{cases} P_r(\mathbf{x}) &= \int_{\mathbf{x}'} \kappa(\mathbf{x}', \mathbf{x}; \sigma) W_r(\mathbf{x}') d\mathbf{x}', \\ P_p(\mathbf{p}) &= \int_{\mathbf{p}'} \kappa(\mathbf{p}', \mathbf{p}; \sigma) W_r(\mathbf{p}') d\mathbf{p}', \end{cases} \tag{3}$$

where $\mathbf{x}'$ is a pixel in the image domain and $\mathbf{p}'$ is a point in the 3D space, $\kappa(\cdot, \cdot; \sigma)$ denotes the gaussian kernel with bandwidth $\sigma$; $P_r(\mathbf{x})$ and $P_p(\mathbf{p})$ denotes the probabilistic density function at pixel $\mathbf{x}$ and point $\mathbf{p}$ respectively.

## 3.2 Geometric Regularization on Implicit Surfaces

Regularizing geometric surface properties is critical to achieving desirable shapes, especially in unconstrained regions. While such constraints can be easily realized with explicit shape representations, a controlled regularization of an implicit surface is not straightforward, since the surface is implicitly encoded as the level set of a scalar field. Here, we introduce a general formulation of geometric regularization for implicit surfaces using a new importance weighting scheme.

Since computing geometric properties of a surface, e.g. normal, curvature, etc., requires access to the derivatives of the field, we propose a finite difference method-based approach. In particular, we

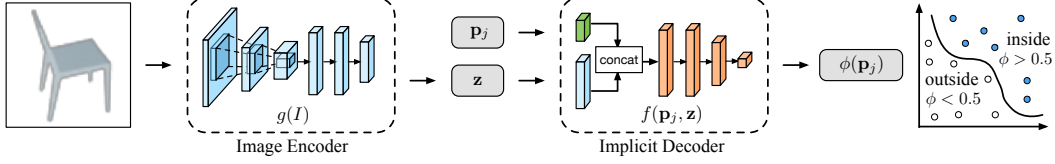

Figure 3: Network architecture for unsupervised learning of implicit surfaces. The input image $I$ is first mapped to a latent feature $\mathbf{z}$ by an image encoder $g$ while the implicit decoder $f$ consumes both the latent code $\mathbf{z}$ and a query point $\mathbf{p}_j$ and predicts its occupancy probability $\phi(\mathbf{p}_j)$. With a trained network, one can generate an implicit field whose iso-surface at 0.5 depicts the inferred geometry.

compute the $n$-order derivative of the implicit field at point $\mathbf{p}_j$ with central difference approximation:

$$\frac{\delta^n \phi}{\delta \mathbf{p}_j^n} = \frac{1}{\Delta d^n} \sum_{l=0}^{n} (-1)^l \binom{n}{l} \phi(\mathbf{p}_j + (\frac{n}{2} - l)\Delta d), \tag{4}$$

where $\Delta d$ is the spacing distance between $\mathbf{p}_j$ and its adjacent sample points (Figure 4). When $n$ equals to 1, the surface normal $\mathbf{n}(\mathbf{p}_j)$ at $\mathbf{p}_j$ can be obtained via $\mathbf{n}(\mathbf{p}_j) = \frac{\delta \phi}{\delta \mathbf{p}_j} / \left| \frac{\delta \phi}{\delta \mathbf{p}_j} \right|$.

**Importance weighting.** As we focus on the geometric properties on the surface, applying the regularizer over the entire 3D space would lead to overly loose constraint in regions of interest. Hence, we propose an importance weighting approach to assign more attention on the sampling points closer to the surface. Here, we leverage the prior learned by our network – the surface points should have an occupancy probability close to the decision boundary, which is 0.5 in our implementation. Therefore, we propose a weighting function $W(x) = \mathbb{I}(|x - 0.5| < \epsilon)$ and formulate the loss of geometric regularization as follows:

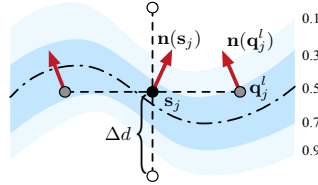

Figure 4: 2D illustration of importance weighted geometric regularization.

$$\mathcal{L}_{geo} = \frac{1}{N_p} \sum_{j=1}^{N_p} W(\phi(\mathbf{s}_j)) \frac{\sum_{l=1}^{6} W(\phi(\mathbf{q}_j^l)) \|\mathbf{n}(\mathbf{s}_j) - \mathbf{n}(\mathbf{q}_j^l)\|_p^p}{\sum_{l=1}^{6} W(\phi(\mathbf{q}_j^l))}. \tag{5}$$

In particular, as shown in Figure 4, for each anchor point $\mathbf{s}_j$, we uniformly sample 2 neighboring samples $\{\mathbf{q}_j^l\}$ with spacing $\Delta d$ along the $x$, $y$ and $z$ axis respectively. We feed the weight function $W(\cdot)$ with the predicted occupancy probability $\phi(\mathbf{s}_j)$ such that anchor points closer to the surface (with $\phi(\mathbf{s}_j)$ closer to 0.5) would receive higher weights and vice versa. By minimizing $\mathcal{L}_{geo}$, we encourage the normals at the 3D anchors to stay close to that of its adjacent points. Notice that we use $l_p$ norm rather than the commonly used $l_2$ for generality. We show that various geometric properties can be achieved by taking $p$ as a hyper parameter (see Section 4.3).

The total loss for network training is a weighted sum of the silhouette loss $\mathcal{L}_{sil}$ and the geometric regularization loss $\mathcal{L}_{geo}$ with a trade-off factor $\lambda$ as shown below:

$$\mathcal{L} = \mathcal{L}_{sil} + \lambda \mathcal{L}_{geo}. \tag{6}$$

# 4 Experiments

## 4.1 Experimental Setup

**Datasets.** We evaluate our method on ShapeNet [10] dataset. We focus on 6 commonly used categories with complex topologies: *plane*, *bench*, *table*, *car*, *chair* and *boat*. We use the same train/validate/test split as in [4, 1, 2] and the rendered images ($64 \times 64$ resolution) provided by [1] which consist of 24 views for each object.

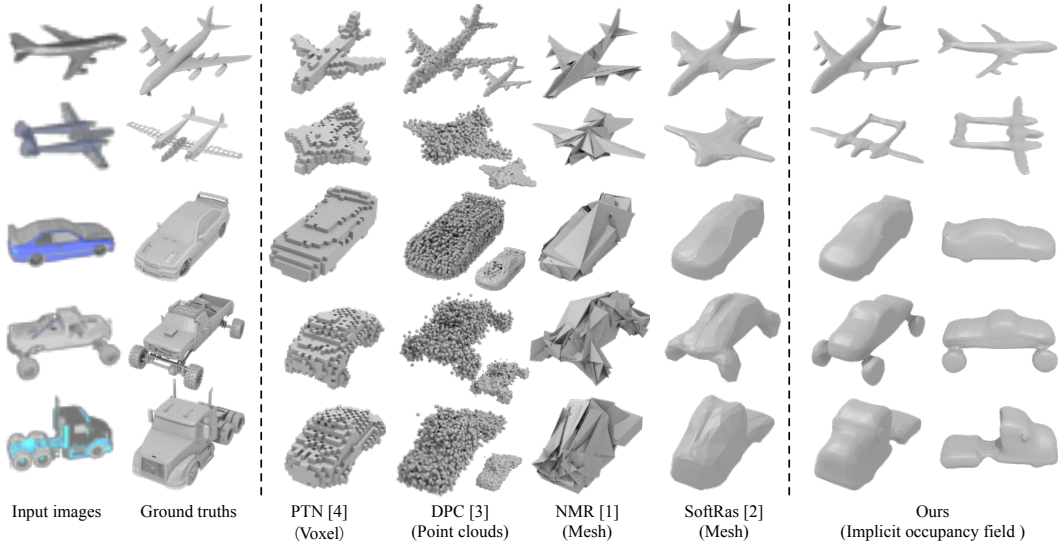

| Input images | Ground truths | PTN [4] (Voxel) | DPC [3] (Point clouds) | NMR [1] (Mesh) | SoftRas [2] (Mesh) | Ours (Implicit occupancy field ) |

Figure 5: Qualitative results of single-view reconstruction using different surface representations. For point cloud representation, we also visualize the meshes reconstructed from the output point cloud.

**Implementation details.** We adopt a pre-trained ResNet18 as the encoder, which outputs a latent code of 128 dimensions. The decoder is realized using 6 fully-connected layers (output channels as 2048, 1024, 512, 256, 128 and 1 respectively) followed by a *sigmoid* activation function. We sample $N_p = 16,000$ anchor points in 3D space and $N_r = 4096$ rays for each view. The sampling bandwidth $\sigma$ is set as $7 \times 10^{-3}$. The radius $\tau$ of the supporting region is set as $3 \times 10^{-2}$. For the regularizer, we set $\Delta d = 3 \times 10^{-2}$, $\lambda = 1 \times 10^{-2}$, and norm $p = 0.8$. We train the network using Adam optimizer with learning rate of $1 \times 10^{-4}$ and batch size of 8 on a single 1080Ti GPU.

## 4.2 Comparisons

We validate the effectiveness of our framework in the task of unsupervised shape digitization from a single image. Figure 5 and Table 1 compare the performance of our approach with the state-of-the-art unsupervised methods that are based on explicit surface representations, including voxels [4], point clouds [3], and triangle meshes [1, 2]. We provide both qualitative and quantitative measures. Note that all the methods are trained with the same training data for fair comparison. While the explicit surface representations either suffer from visually unpleasant reconstruction due to limited resolution and expressiveness (voxels, point clouds), or fail to capture complex topology from a single template (meshes), our approach produces visually appealing reconstructions for complex shapes with arbitrary topologies. Compared to mesh-based representations, we are able to achieve higher resolution output, which is reflected by the even sharper local geometric details, e.g. the engine of plane (first row) and the wheels of the vehicle (fourth row). The performance of our method is also demonstrated in the quantitative comparisons, where we achieve state-of-the-art reconstruction accuracy using 3D IoU with large margins.

In Figure 6, we further illustrate the importance of supporting arbitrary topologies, compared to existing mesh-based reconstruction techniques [2]. Since real-world objects can exhibit a wide range of varying topologies even for a single object category (e.g., chairs), mesh-based approaches often lead to deteriorated results. In contrast, our approach is able to faithfully infer complex shapes and arbitrary topologies from very limited visual cues, e.g. the chair and the table on the third row, thanks to the flexibility of the implicit representation and the strong shape prior enabled through the geometric regularizer.

## 4.3 Ablation Analysis

We provide a comprehensive ablation study to assess the effectiveness of each algorithmic component. For all the experiments, we use the same data and parameters as before unless otherwise noted.

| Category | Airplane | Bench | Table | Car | Chair | Boat | Mean |
|---|---|---|---|---|---|---|---|
| PTN [4] | 0.5564 | 0.4875 | 0.4938 | 0.7123 | 0.4494 | 0.5507 | 0.5417 |
| NMR [1] | 0.6172 | 0.4998 | 0.4829 | 0.7095 | 0.4990 | 0.5953 | 0.5673 |
| SoftRas [2] | 0.6419 | 0.5080 | 0.4487 | 0.7697 | 0.5270 | **0.6145** | 0.5850 |
| Ours | **0.6510** | **0.5360** | **0.5150** | **0.7820** | **0.5480** | 0.6080 | **0.6067** |

Table 1: Comparison of 3D IoU with other unsupervised reconstruction methods.

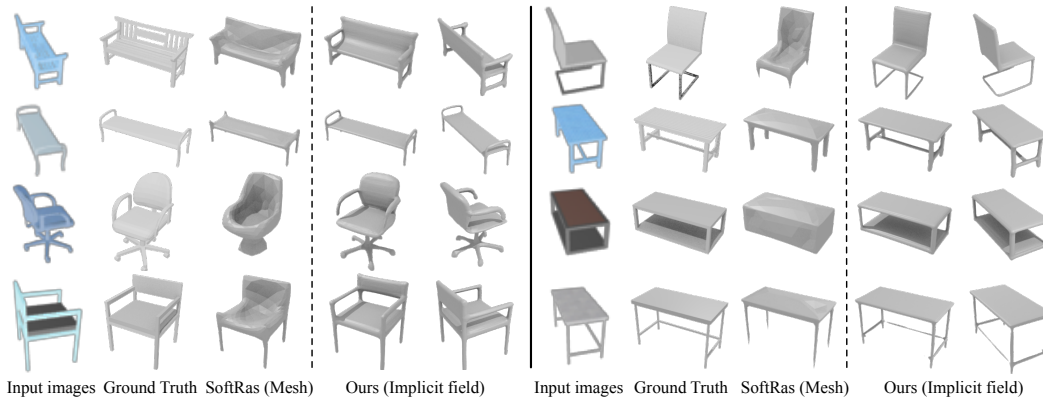

Input images  Ground Truth  SoftRas (Mesh)  Ours (Implicit field)  Input images  Ground Truth  SoftRas (Mesh)  Ours (Implicit field)

Figure 6: Qualitative comparisons with mesh-based approach [2] in term of modeling capability of capturing varying topologies.

**Geometric Regularization.** In Table 2 and Figure 7, we demonstrate that our proposed geometric regularization enables a flexible control over various geometric properties by varying the value of norm $p$. To validate the effectiveness of geometric regularization, we train the same network using different configurations: 1) without using any geometry regularizers; 2) applying our proposed geometric regularization with $p$ norm equals to 0.8, 1.0, 2.0, respectively. As shown in the results, the lack of geometry regularizer would lead to an ambiguity of reconstructed geometry, e.g. first row in Figure 7, as some unexpected shape could appear the same with the ground-truth with an accordingly optimized texture map, and thus makes the generation of flat surface rather difficult. The proposed regularizer can effectively enhance the regularity of reconstructed objects, especially for man-made objects, while providing flexible control. In particular, when $p = 2.0$, the surface normal difference is minimized in a least-square manner, leading to a smooth reconstruction. When $p \to 0$, sparsity is enforced in the surface normal consistency, which encourages the reconstructed surface to be piece-wise linear and is often desirable for man-made objects. We also perform ablation study on the effect of the sampling step $\Delta d$ for the regularizer as shown in Table 3 and Figure 8. We can observe that larger $\Delta d$ leads to more flattening surfaces at the cost of less fine details.

| 3D IoU | |
|---|---|
| norm p = 2.0 | 0.502 |
| norm p = 1.0 | 0.524 |
| norm p = 0.8 | **0.548** |
| -Regularizer | 0.503 |

Table 2: Quantitative evaluations of our approach on chair category using different regularizer configurations.

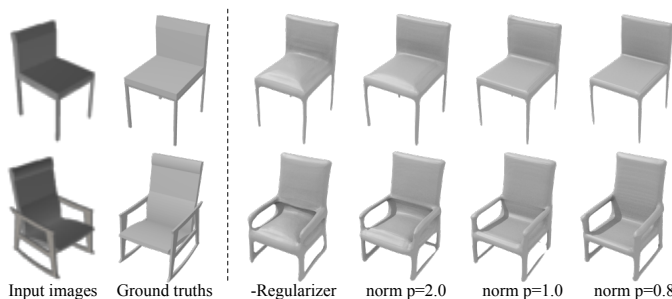

Input images  Ground truths  -Regularizer  norm p=2.0  norm p=1.0  norm p=0.8

Figure 7: Qualitative evaluations of geometric regularization by using different configurations.

**Importance Sampling.** To fully explore the effect of importance sampling, we compare two different configurations of sampling scheme: 1) "-Imp. sampling": drawing both 3D anchor points and rays from the normal distribution with mean and standard deviation set as 0 and 0.4 respectively; and 2) "Full model": only using the importance sampling approach for both anchor points and rays with the bandwidth set as 0.007. We show sampled rays and results in Table 4 and Figure 9. In terms of

visual quality, importance sampling based approach has achieved much more detailed reconstruction compared to its counterpart. The quantitative measurement also leads to consistent observation, where our proposed importance sampling has outperformed the normal sampling by a large margin.

| 3D IoU | |
|---|---|
| $\Delta d = 1 \times 10^{-2}$ | 0.482 |
| $\Delta d = 3 \times 10^{-2}$ | **0.515** |
| $\Delta d = 1 \times 10^{-1}$ | 0.507 |

Table 3: Quantitative evaluations on table category with different $\Delta d$

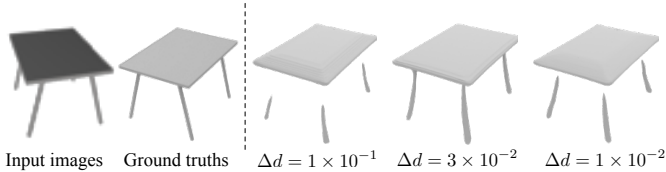

Figure 8: Qualitative results of reconstruction using our approach with different regularizer sampling step $\Delta d$.

| 3D IoU | |
|---|---|
| Full model | **0.548** |
| -Imp. sampling | 0.482 |
| -Boundary aware | 0.524 |

Table 4: Quantitative measurements for the ablation analysis of importance sampling and boundary-aware assignment on the chair category as shown in Figure 9.

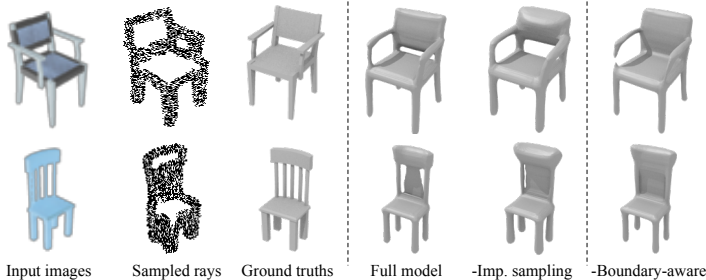

Figure 9: Qualitative analysis of importance sampling and boundary-aware assignment for single-view reconstruction.

**Boundary-Aware Assignment.** We also compare the performance with and without boundary-aware assignment in Table 4 and Figure 9. When boundary-aware assignment is disabled, the sampling rays around the decision boundary may be assigned with incorrect labels. As a result, the reconstructions lack sufficient accuracy, especially around the thin surface regions, and thus may not be able to capture holes and thin structures as demonstrated in the rightmost examples in Figure 9.

## 5  Discussion

We introduced a learning framework for implicit surface modeling of general objects without 3D supervision. An occupancy field is learned through a set of 2D silhouettes using an efficient field probing algorithm, and the desired local smoothness of implicit field is achieved using a novel geometric regularizer based on finite difference. Our experiments show that high-fidelity implicit surface modeling is possible from 2D images alone, even for unconstrained regions. Our approach can produce more visually pleasant and higher-resolution results compared to both voxels and point clouds. In addition, unlike mesh representations, our approach can handle arbitrary topologies spanning various object categories. We believe that the use of implicit surfaces and our proposed algorithms opens up new frontiers for learning limitless shape variations from in-the-wild images. Future work includes unsupervised learning of textured geometries, which has been recently addressed with an explicit mesh representation [2], and eliminating the need of silhouette segmentations to further increase the scalability of the image-based learning. It would also be interesting to investigate the use of anisotropic kernels for shape modeling and hierarchical implicit representations with advanced data structure, e.g. Octree, to further improve the modeling efficiency. Furthermore, we would like to consider the use of learning from texture cues in addition to binary masks.

**Acknowledgements**   This research was conducted at USC and was funded by in part by the ONR YIP grant N00014-17-S-FO14, the CONIX Research Center, a Semiconductor Research Corporation (SRC) program sponsored by DARPA, the Andrew and Erna Viterbi Early Career Chair, the U.S. Army Research Laboratory (ARL) under contract number W911NF-14-D-0005, Adobe, and Sony.

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
