[Supplementary Material]

# Learning to Infer Implicit Surfaces without 3D Supervision – Supplemental Materials

## 1 More Results

We show more results of our approach in Figure 1 and Figure 2.

Figure 1: More single-view reconstruction results of our approach on bench and car categories.

Input images    Ground truth    Ours (6 views)

Figure 2: More single-view reconstruction results of our approach on chair, plane and table categories.