[Reviews · NeurIPS 2019]

Reviewer 1



Even though the method needs 2D supervision, all experiments and visualizations seems to suggest that approach needs images with single objects with perfectly segmented transparent/white background. All the experiments were done by synthetic renderings of CAD models itself. So in practice there is no benefit over 3D supervision from CAD models. From the experiments you have to take a big leap of faith to assume that the approach works with occlusions and imperfect masks (say from MaskRCNN like systems). Not convinced from the ablation evaluation that the geometric regularization is helping. In terms of 3D IoU there is no statistically significant improvement in 3D IoU scores of 0.503 vs 0.502. I would like to see ablation study done across multiple shape categories and increasing amount of training data. I wonder with more data (since the proposed method only needs 2D supervision) if these hand engineered 3D regularizations have any benefit? Equation1 seems to be an approximation of the volume rendering integral? Can you explain the assumptions made there and place it with respect to the computer graphics literature on volume rendering. Overall the paper is well-written and easy to follow. However some section needs improvement. Fig2 illustration is not clear until one reads the “Boundary Aware Assignment” paragraph (lines 168-178). Refering to that paragraph in the Figure2 caption can help with the clarity. Figure3 is excellent and very clearly depicts the network architecture. In my opinion, it should be before Figure2. Please improve Table2 headers. This is the most important quantitative study in the paper describing the ablation of each component in the paper. This is a fast moving field, but I think these are highly relevant related works that the authors missed. DeepSDF (Park et. al. 2019) and Occupancy Networks (Mescheder et al. 2019) both uses a similar implicit representation and a shape decoder conditioned on 3D location. 2D supervision from masks for predicting shapes as TSDF (implicit) representation has been done before (3D-RCNN Kundu et. al. 2018) but not in an end-to-end fashion. Why choose to have the implicit values in the range of [0, 1] as occupancy probability instead of having zero-level set. Isn’t it is beneficial to have a zero-mean prediction like in a standard Signed Distance Function (SDF)? Also if the implicit representation in this paper is indeed an occupancy probability, then Differentiable Ray Consistency (Tulsiani et. al. 2017) has already demonstrated how to backpropagate through occupancy values collected on a ray for single view 3D shape learning.

Reviewer 2



This paper addresses the unsupervised 3D shape generation problem by using the implicit surface function. The paper is well written and easy to follow. My biggest concern is that while the experiments show the implicit function can achieve better results than explicit representations, the training time and inference time are not shown and compared. And how the authors infer the 3D shape at evaluation and what resolution they adopt can be described in more details. Another question is how the support region radius affects the prediction.

Reviewer 3



The idea makes completely sense, the paper introduces a couple of techniques to make it work. The results are far better than the state-of-the-art in other representations. The paper is well-written. Other comments: 1) Would it be better to use some anisotropic kernels to capture shape structures? 2) Can we make the implicit representation hierarchical? 3) It would be also interesting to see how the results change when changing the viewpoints of the input image? 4) What does equation (6) approximate in the limit? Curvature?

[Author Response · NeurIPS 2019]

We thank the reviewers for the valuable feedback. While we address the major concerns below, we will include and discuss the missing references (R1, R3), improve and clarify the captions of Figure 2 and Table 2 (R1), and move Figure 3 before Figure 2 (R1) in the revised version.

**(R1) Q: Requirement of perfect segmentation and occlusion handling. A:** While obtaining accurate 3D annotations remains a challenging task even for humans, accurate 2D segmentation has become more accessible by either state-of-the-art instance segmentation methods or semi-automatic image processing software. Hence, we believe that learning 3D shapes from 2D supervision can be a much more practical solution for scalable 3D reconstruction. We agree that learning shape representations in the presence of occlusions or imperfect masks remains challenging and it would be a very interesting direction. We will add the above discussion to the limitations and the future work in the revision.

**(R1) Q: Marginal improvement of geometric regularization (0.502 vs 0.503). A:** As described in the implementation details (line 231-232), we use $p = 0.8$ as our final model. The ablation study demonstrates that using the geometric regularization with $p = 0.8$ significantly improves 3D IOU **from 0.503 to 0.554**. We will highlight the ablation choices in Table 2 to clarify the improvement. In addition, we will provide results with more categories and input viewpoints in the next revision to provide additional evaluations.

**(R1) Q: Assumption and background of Equation 1. A:** By viewing the occupancy probability as a thickness of a translucent medium [36], we compute the occupancy probability on the image plane by integrating the multiplication of the transmittance (i.e., visibility) and occupancy probability at any point along the ray as denoted in Equation 1. Also the visibility of a point along the ray is exponentially decayed with the accumulated density. The same formulation has been used in computer vision literature for differentiable approximation of 3D scene [Rhodin et al. 2015]. We are happy to elaborate more on the formulation in the revision.

**(R1) Q: Why implicit value in [0, 1] instead of zero-level set? A:** Without 3D ground truth, it is not possible to obtain the signed distance field in 3D. Thus, we use an occupancy field from $[0, 1]$ which is also used in state-of-the-art 3D reconstruction methods, e.g. Occupancy Network [8], where $0.5$ is set as decis ion boundary. In addition, our approach uses 0.5 with sigmoid function, which is equivalent to the common zero-mean prediction with tanh activation.

**(R1) Q: Difference with the Differentiable Ray Consistency (DRC) paper. A:** In DRC, the occupancy fields are stored in explicit voxel grids ($32^3$), where uniform voxel-wise sampling along ray suffices to propagate gradients to all the voxel grids. In contrast, an implicit shape representation encodes geometry with unlimited resolution at the cost of a dense evaluation of the implicit function, which is not tractable with the uniform sampling and aggregation method proposed in DRC. Hence, we propose an efficient sampling approach for differentiable ray marching with implicit surfaces based on importance, which is one of our main contributions.

**(R1) Q: Speed improvement of field probing approach. A:** A naive solution of densely evaluating all points in resolution $R_v^3$ and aggregating them into pixels in resolution $R_p^2$ leads to a computational cost of $O(R_v^3)$ and $O(R_p^2 R_v^3)$ for network forwarding and intersection computation, respectively. Such cost is typically intractable for network training with commodity GPUs. Our field probing approach largely reduces the computational complexity to $O(M_v M_p)$ where $M_v$ and $M_p$ stand for the number of 3D anchor points and 2D image pixels respectively with importance sampling. Our experiments show that our method is two magnitudes faster than the naive approach, making it feasible to train a shape inference network with substantially higher resolution.

**(R2) Q: Training and inference time; evaluation method and resolution. A:** For each category, our model is trained for one day with a single RTX 2080 Ti GPU. It takes 2s to infer each object at a resolution of $256^3$. For evaluation, we reconstruct the geometry by densely evaluating the implicit function at a regular 3D grid with resolution $256^3$. To compute the 3D IoU, we downsample the voxels to $32^3$ using max pooling and compare it with the ground truth. We will include these details in the revision.

**(R2) Q: How does the support region radius affect the prediction? A:** A smaller radius of support region helps to reconstruct finer details at the cost of sampling denser points and rays. We found that setting the radius $\tau$ as the average distance between any two nearest adjacent anchor points ($\tau = 0.03$) strikes the best trade-off between the prediction accuracy and computational cost with 3D IOU of $0.554$. Using the same training parameters except the radius, we obtain 3D IOU of $0.515$ and $0.502$ with radius of $0.01$ and $0.05$ respectively on a chair category. We observed that training with a larger radius tends to ignore details and the one with a smaller radius suffers from insufficient ray assignment. We will add an ablation study of the support region radius in the revision.

**(R3) Q: Using anisotropic kernels for shape modeling? A:** We found it non-trivial to use anisotropic kernels for implicit shape learning as it introduces additional parameters to either set or learn (i.e., radius in each axis and axis rotation). We are happy to discuss this as part of future work.

**(R3) Q: About hierarchical implicit representation. A:** We believe that our method can be extended to learn hierarchical implicit representations using a spatial data structure such as Octrees and will mention this in the future work section.

**(R3) Q: Effect of changing input viewpoints A:** We found that our prediction is reasonably consistent when changing view points. Note that our model is trained with multiview inputs and the numerical evaluations use all $24$ views, hence they are not biased w.r.t. views. We will provide more results using different viewpoints to better evaluate our approach.

**(R3) Q: What does Equation 6 approximate in the limit? A:** We believe R3 meant Equation 5. Our intention of Equation 5 is to encourage local smoothness and the effective smoothed region is controlled by the perturbation $\Delta d$. Hence, at the limit, the equation would lead to smoothness for an infinitesimally small volume.

**(R3) Q: Additional graphics papers to be discussed. A:** We will include and discuss the suggested papers in the revision.

[Meta-Review · NeurIPS 2019]

This paper received uniformly very positive reviews. All reviewers agreed that developing 3D vision systems that do not rely on 3D training data is an extremely important problem, that implicit surfaces are a promising representation, and that this paper proposes a useful technique that could inspire future work.